BCD Beam Search: considering suboptimal partial solutions in Bad Clade Deletion supertrees

Fleischauer Markus markus.fleischauer@uni-jena.de
http://orcid.org/0000-0002-9304-8091 Böcker Sebastian
Chair for Bioinformatics, Friedrich-Schiller-University , Jena , Germany
Hao Jin-Kao
Electronic publication date: 2018 Jun 8
Publication date: 2018
Volume: 6
Electronic Location ID: e4987
Received 2018 Mar 11; Accepted 2018 May 26
Copyright: © 2018 Fleischauer and Böcker
Copyright year: 2018
Copyright holder: Fleischauer and Böcker
License: This is an open access article distributed under the terms of the Creative Commons Attribution License, which permits unrestricted use, distribution, reproduction and adaptation in any medium and for any purpose provided that it is properly attributed. For attribution, the original author(s), title, publication source (PeerJ) and either DOI or URL of the article must be cited.
License URL: https://creativecommons.org/licenses/by/4.0/

Keywords: MRC, Matrix representation with parsimony, Split fit, Phylogeny, Phylogenetics, Supermatrix, Supertree, Divide-and-conquer, Minimum cut

Funding: Deutsche Forschungsgemeinschaft BO∼1910/12 This work was supported by the Deutsche Forschungsgemeinschaft, project BO∼1910/12. The funders had no role in study design, data collection and analysis, decision to publish, or preparation of the manuscript.

==============================
Supertree methods enable the reconstruction of large phylogenies. The supertree problem can be formalized in different ways in order to cope with contradictory information in the input. Some supertree methods are based on encoding the input trees in a matrix; other methods try to find minimum cuts in some graph. Recently, we introduced Bad Clade Deletion (BCD) supertrees which combines the graph-based computation of minimum cuts with optimizing a global objective function on the matrix representation of the input trees. The BCD supertree method has guaranteed polynomial running time and is very swift in practice. The quality of reconstructed supertrees was superior to matrix representation with parsimony (MRP) and usually on par with SuperFine for simulated data; but particularly for biological data, quality of BCD supertrees could not keep up with SuperFine supertrees. Here, we present a beam search extension for the BCD algorithm that keeps alive a constant number of partial solutions in each top-down iteration phase. The guaranteed worst-case running time of the new algorithm is still polynomial in the size of the input. We present an exact and a randomized subroutine to generate suboptimal partial solutions. Both beam search approaches consistently improve supertree quality on all evaluated datasets when keeping 25 suboptimal solutions alive. Supertree quality of the BCD Beam Search algorithm is on par with MRP and SuperFine even for biological data. This is the best performance of a polynomial-time supertree algorithm reported so far.

Introduction

Supertree methods assemble phylogenetic trees with non-identical but overlapping taxon sets into a larger supertree that contains all taxa of each input tree. Constructing a rooted supertree is easy if no contradictory information is encoded in the input trees (Aho et al., 1981); the difficulty stems from resolving conflicts in a reasonable and swift way. Many supertree methods have been proposed over the years; see Bininda-Emonds (2004) for early methods, and Ross & Rodrigo (2004), Criscuolo et al. (2006), Chen et al. (2006), Holland et al. (2007), Cotton & Wilkinson (2007), Steel & Rodrigo (2008), Scornavacca et al. (2008), Ranwez, Criscuolo & Douzery (2010), Bansal et al. (2010), Snir & Rao (2010), McMorris & Wilkinson (2011), Swenson et al. (2012), Brinkmeyer, Griebel & Böcker (2013), Berry, Bininda-Emonds & Semple (2013),Vachaspati & Warnow (2016), Markin & Eulenstein (2016) and Fleischauer & Böcker (2017) for recent ones. Conflicts can be caused by estimation errors during source tree computation, or by evolutionary processes (e.g., incomplete lineage sorting or horizontal gene transfer) resulting in conflicting gene trees. The latter problem is known as Gene Tree Species Tree Reconciliation problem; for this problem, methods were developed that incorporate evolutionary processes such as the coalescent process (Liu et al., 2009; Larget et al., 2010; Liu, Yu & Edwards, 2010; Liu & Yu, 2011; Whidden, Zeh & Beiko, 2014; Mirarab et al., 2014; Allman, Degnan & Rhodes, 2016). Reconciliation approaches model the evolutionary process more thoroughly than standard supertree methods, but do not scale well with the number of taxa. Supertree methods are complemented by supermatrix methods, which do not combine the trees but rather the “raw” sequence data (von Haeseler, 2012).

To build large-scale phylogenies, a promising approach is to use supertree methods as part of divide-and-conquer meta techniques, as pioneered by the disk-covering method (Huson, Nettles & Warnow, 1999a; Huson, Vawter & Warnow, 1999b; Roshan et al., 2004) which was further developed to DACTAL (Nelesen et al., 2012). We break down a large phylogenetic problem into smaller and easier-to-solve subproblems. Conflicts between subproblem solutions will mainly result from sampling errors, which qualifies supertree methods to combine the sub-solutions. With the usage of fast (polynomial-time) and accurate supertree methods, a divide-and-conquer strategy promises gains in both accuracy and speed compared to a conventional phylogenetic analysis.

Matrix representation (MR) supertree methods encode inner nodes of all input trees as partial binary characters in a matrix, which is then analyzed using an optimization or agreement criterion to yield the supertree. Matrix representation with parsimony (MRP) (Baum, 1992; Ragan, 1992) is still the most widely used supertree method today. Constructed supertrees are of comparatively high quality and, until recently, no supertree method consistently outperformed MRP on datasets with 500+ taxa (Swenson et al., 2010, 2011; Brinkmeyer, Griebel & Böcker, 2011). MRP is NP-hard (Foulds & Graham, 1982), so heuristic search strategies have to be employed. Swenson et al. (2012) introduced the meta-method SuperFine which combines the greedy strict consensus merger (GSCM) (Huson, Vawter & Warnow, 1999b) with MRP. SuperFine outperformed all other methods, including MRP, on a variety of simulated and biological datasets. SuperFine using supertree methods other than MRP, did not result in improved supertree quality (Swenson et al., 2012; Nguyen, Mirarab & Warnow, 2012). MRP is complemented by matrix representation with flipping (MRF) (Burleigh et al., 2004) and matrix representation with compatibility (MRC) (Purvis, 1995; Rodrigo, 1996; Ross & Rodrigo, 2004; Creevey & Mcinerney, 2005). MRF seeks the minimum number of “flips” that make the resulting matrix compatible. Brinkmeyer, Griebel & Böcker (2013) introduced the FlipCut supertree method as a very swift polynomial-time heuristic for the MRF problem. The accuracy of FlipCut was superior compared to other polynomial-time supertree methods, but still worse and less robust than MRP. MRC searches for the maximal subset or matrix columns that are pairwise compatible (maximum independent set). Recently, Fleischauer & Böcker (2017) introduced Bad Clade Deletion (BCD) supertrees, which adopts the FlipCut idea but uses a different objective function: namely, Minimum Column Deletion (MCD) which minimizes the number of matrix columns that have to be deleted to resolve all conflicts in the input matrix (minimum vertex cover). MRF, MRC and MCD are again NP-complete (Böcker et al., 2011); MCD and MRC are complementary problems, but differ with respect to approximability and parameterized complexity. BCD performs on par with or even better than the established supertree methods SuperFine and MRP. It inherits the guaranteed polynomial running time from FlipCut and is even faster in practice. But the greedy search strategy performed by BCD struggles to find a close-to-optimum solution on highly conflicting data when no or unreliable meta information (e.g., bootstrap (BS) values or branch lengths (BLs)) is available (Fleischauer & Böcker, 2017).

Here, we present a beam search approach for BCD, to consider not only the best but the k best solutions in every phase of the top-down construction of the supertree. We introduce and evaluate an exact and a randomized subroutine to calculate suboptimal solutions. Both variants still have guaranteed polynomial running time, a highly desirable property in the context of divide-and-conquer tree reconstruction. In our evaluation on multiple simulated and biological datasets, we found that both beam search approaches consistently outperform “classical” BCD supertrees. The randomized approach performs only negligibly worse than the exact enumeration, but has only a linear instead of a quadratic running time dependency on the number of suboptimal solutions.

Preliminaries

In this paper, we deal with three types of graph-theoretical objects: namely, phylogenetic trees, graphs that we search for cuts and vertex-cuts, and networks that we search for cuts and flows. In contrast to a cut, a vertex-cut is the partition of a connected graph that is induced by vertex deletions instead of edge deletions. For readability, vertices of a tree will be called nodes, whereas directed edges of a network will be referred to as arcs.

Phylogenetic trees

Let n be the number of taxa in our study; for brevity, we assume that our set of taxa equals {1, …, n}. In this paper, we assume all trees to be rooted phylogenetic trees. We call all nodes with out-degree zero leaves, and all non-leaf nodes except the root internal nodes. All leaves of the trees are (labeled with) taxa from {1, …, n}, no taxon appears twice in a tree, and there exist no nodes with out-degree one. Given a set of input trees T1, …, Tl with leaf set ℒ(Ti)⊆{1,…,n}, we assume ∪i ℒ(Ti)={1,…,n}. We search for a supertree T of these input trees, that is, a tree with leaf set ℒ(T)={1,…,n}. For Y⊆ℒ(T), we define the Y-induced subtree T|Y of T as the minimal induced subgraph of T that connects Y, where we contract the outgoing edge of each internal node with out-degree one. To contract an edge (v, c), we replace (v, c) and the incoming edge (p, v) by (p, c), and remove v. Some tree T refines T′ if T′ can be reached from T by contracting internal edges. We say that a supertree T of T1, …, Tl is a parent tree if T|ℒ(Ti) refines Ti, for all i = 1, …, l.

Matrix representation of phylogenetic trees

We can encode a tree T with taxon set {1, …, n} in the input trees in a matrix M(T) with elements in {0, 1}: Each row of the matrix corresponds to one taxon. For simplicity we assume that there is some natural ordering of the taxa and, hence, the rows. Each inner node except the root is called a clade and encoded in one column of the matrix M(T): Entry “1” indicates that the corresponding taxon is a leaf of the subtree rooted in the clade, whereas all other taxa are encoded “0”. For trees T, T′ with identical taxa, T refines T′ if and only if M(T′) can be obtained from M(T) by column deletion (Brinkmeyer, Griebel & Böcker, 2013). According to the classical perfect phylogeny model (Wilson, 1965), a binary matrix M admits a perfect phylogeny if there is a rooted tree with n leaves corresponding to the n taxa, such that there is a one-to-one correspondence between matrix columns and clades of the tree: For column u there exists a node u′ of the tree such that M = 1 holds if and only if taxon t is a leaf of the subtree below u′ for all t, and vice versa. One important characterization of a matrix to admit a perfect phylogeny, is that any two columns of the matrix must be compatible: Let A be the set of all taxa with entry “1” in the first column, and let B be the set of all taxa with entry “1” in the second column, then A⊆B or A⊇B or A∩B=∅ must hold.

We generalize the matrix encoding for a set of trees T={T1,…,Tl} with taxon sets ℒ(Ti)⊆{1,…,n}: We encode the input trees in a matrix M(𝒯) with elements in {0, 1, ?}. Again, each row of the matrix corresponds to one taxon. Each non-trivial clade in each input tree is encoded in one column of the matrix M(𝒯): The entries for taxa t∈ℒ(Ti) are set to either “1” or “0” as defined above, whereas the state of taxa that are not part of Ti is unknown, and represented by a question mark (“?”). We assume that the set of matrix columns equals {1, …, m}, then clearly m ≤ l(n−2). In detail, m is the total number of non-trivial clades in T1, …, Tl. The matrix M(𝒯) has size m × n and can be computed in O(mn) time, using a tree traversal and lists of taxa. Now, a collection of trees 𝒯 has a parent tree if and only if M(𝒯) is an incomplete (directed) perfect phylogeny (Brinkmeyer, Griebel & Böcker, 2013; Pe’er et al., 2004).

The Bad Clade Deletion algorithm

To resolve incompatibilities among the input trees, the MCD model assumes that the matrix M is perturbed. We search for a perfect phylogeny matrix M* such that the number of matrix columns we have to delete is minimal. The following construction is due to (Pe’er et al., 2004): For a subset S⊆{1,…,n} of taxa and a subset D⊆{1,…,m} of clades, G(S, D) is a bipartite graph with the vertex sets S and D; there is an edge between t ∈ S and c ∈ D if and only if M[t, c] = 1 holds. A clade vertex c ∈ D is semiuniversal (in S, D) if M[t, c] ∈ {1, ?} holds for all t ∈ S. We remove all semiuniversal clade vertices from the graph, as all “?”-entries can be resolved to “1” (Pe’er et al., 2004).

The BCD algorithm proceeds as follows: For S←{1,…,n} and D←{1,…,m}, we construct the graph G(S, D) as defined above. If G(S, D) is disconnected, we recursively repeat the above construction for each connected component S′, D′ with |S′|>1. The subsets S′ of taxa computed during all recursion steps of the algorithm, form a hierarchy that represents the BCD supertree. But if G(S, D) is connected at some stage, we disconnect the set of taxa S by searching for a vertex-cut in G(S, D). Removing clade vertices from G(S, D) is equivalent to deleting the corresponding matrix columns from M. Formally, we assume that each clade vertex c has some weight w(c); the weight of a vertex-cut is ∑c∈U w(c) where U⊆D is the set of clade vertices that has to be removed from D. We search for a vertex-cut of minimal weight.

To efficiently find a minimum vertex-cut of G(S, D), we transform G(S, D) into a directed network H(S, D′) with the same taxon set S. For each clade vertex c ∈ D we create two vertices c− ∈ D′ and c+ ∈ D′ plus an arc (c−, c+) in the network. For each edge (c, t) in G, we insert two arcs (t, c−) and (c+, t) into the network. Arcs (c−, c+) have capacity w(c), whereas all other arcs have infinite capacity. We fix one taxon vertex s, and for all other taxa vertices t we search for a minimum s-t-cut. Among these cuts, the cut with minimal weight is equivalent to a minimum vertex-cut in G(S, D) (Fleischauer & Böcker, 2017). A minimum vertex-cut can be computed in O(mn2) time (Brinkmeyer, Griebel & Böcker, 2013).

The BCD algorithm proceeds in n−1 phases; in each phase, the clade S⊆{1,…,n} is added to the output, and a bipartition of S is computed. The algorithm proceeds greedily, by choosing the best bipartition in every phase. We refer to (Fleischauer & Böcker, 2017; Brinkmeyer, Griebel & Böcker, 2013) for details.

Lemma 1 (Brinkmeyer, Griebel & Böcker, 2013) Given an input matrix M over {0, 1, ?} for n taxa and m clades, the BCD algorithm computes a supertree in O(mn3) time.

By weighting vertices in G(S, D), we can incorporate information about the “reliability of clades” in the source trees: (Fleischauer & Böcker, 2017) introduced BS weighting and BL weighting for this purpose, and found that these weightings significantly improve supertree quality. We only have to use unit weights (UW) if no BS values or BLs are available for the source trees.

Bad Clade Deletion Beam Search

Recall that the BCD algorithm tries to minimize a global objective function: Namely, the total weight of column deletions in the input matrix. Besides the theoretical amenity of this feature, this allows us to compare solutions based on the objective function. But in fact, we can extend this evaluation to partial solutions: At any point of the algorithm, we know the quality of a partial solution, that is, the total weight of clade deletions that were required up to this point. Clearly, this weight only increases during later steps of the algorithm.

Recall that the BCD algorithm proceeds in n−1 phases. An alternative view of the algorithm will be helpful in the following: BCD computes the supertree by iteratively refining a partial phylogenetic tree, which is a phylogenetic tree where several taxa can be mapped to the same leaf. Initially, we have a partial phylogenetic tree with a single node and all taxa attached to it. In each phase, the partial tree is refined by finding a bipartition of the taxa attached to one of the leaves. For the moment, we ignore the case that taxa are partitioned into q > 2 sets, corresponding to polytomies in the supertree, see below. Before phase p, the partial tree has p leaves, partitioning the taxa into p sets.

We will now extend the greedy BCD algorithm by keeping more than one partial solution “alive” in each phase, resulting in a beam search algorithm. The parameter k ≥ 1 determines the number of partial solutions that are considered simultaneously; for k = 1 this is equivalent to the original BCD algorithm. Formally, a partial solution P = (ℳ, T, cost) of order p consists of a set ℳ = {(S1, D1),…,(Sp, Dp)} such that S1, …, Sp is a partition of the taxa S = {1, …, n}, and D1, …, Dp is a partition of the clade vertices D({1, …, m});

a partial phylogenetic tree T with p leaves, labeled by S1, …, Sp; and

a real number cost, the cost for matrix modifications up to this point.

Before phase p of the beam search, we have a set 𝒫 of |𝒫| = k partial solutions of order p. We transform this into a set 𝒫′ of |𝒫′| = k partial solutions of order p + 1. In the first step of the algorithm, we start with a single partial solution with cost zero.

Now, we describe how to transform a partial solution P = (ℳ, T, cost) of order p, into k new partial solutions of order p + 1. For each of the p graphs G(Si, Di) for i = 1, …, p, we compute the k best bipartitions. Out of the resulting pk cuts, we extract the best k cuts in any of the graphs. We iterate over these cuts: Assume that the cut happens in the graph G(S, D) for (S,D) ∈ ℳ. By this cut, both the taxon set S and the clade vertex set D are bipartitioned into sets S′, S″ := S\S′, and D′, D″ :=D\D′. We build a new partial solution P′=(ℳ,T′,cost') as follows: Set ℳ':=ℳ∖{(S,D)}∪{(S′,D′),(S″,D″)};

resolve the node in T labeled S by two nodes S′, S″ in T′;

compute the new costs cost′ := cost + cut where cut are the costs of the cut in G(S, D).

We now evaluate the partial solutions that belong to the same phase p, based on the costs generated so far: In each phase, we do our computations for each of the k partial solutions P ∈ 𝒫. For each partial solution P, we compute k cuts instead of a single one, resulting in k2 partial solutions. We then keep only the best k partial solutions in phase p + 1, each of which is used for computation of cuts in the next phase of the algorithm. See Algorithm 1 for a pseudo code of the algorithm described up to this point.

There is another pitfall we have to consider: In the original algorithm, the order in which we processed the leaves of a partial solution was of no importance, as we eventually had to resolve each leaf. For the beam search, this is no longer the case, as we search for the k best partial solutions. To this end, the algorithm described above computes, for each partial solution in phase p, k cuts in each of the p graphs G(Si, Di) for i = 1, …, p. This would result in an additional O(n) factor in the total running time. But this is in fact not a problem: In each phase, we record all k cuts for each of the p graphs G(Si, Di). For each new partial solution of order p + 1, only one cut in some graph G(S, D) is chosen, whereas all other graphs will reappear unchanged in the next phase. Hence, in the next phase, only two graphs have to be searched for k best cuts: namely, the graphs G(S′, D′) and G(S′, D′). For all other graphs, we already know the k best cuts from the previous phase.

Finally, let us consider the case that a minimum cut results in more than two connected components. This simply means that the resulting partial solution is of higher order than p + 1. When the algorithm is in phase p it will ignore all partial solutions of order above p; in fact, if there exist k′ partial solution s of order above p then we only have to compute k−k′ instead of k cuts. The same holds true if G(S, D) does not require cutting, because it already consists of two or more connected components.

It is understood that BCD Beam Search can be applied for any type of vertex weighting in G(S, D), including BS weights, BL weights, and UWs.

Algorithm 1 BCD Beam Search

1:	function bcdBeamSearch (G(S, D), k)	
2:	  #Taxa ← |S|	
3:	  𝒫 ← initPartialSolutions (G(S, D))	
4:	  while ∃(ℳ, T, cost) ∈ 𝒫:|ℳ| ≤ # Taxa do	
5:	   𝒫′ ← {}	
6:	   for all (ℳ, T, cost) ∈ 𝒫 do	
7:	     for all (S, D) ∈ ℳ do	
8:	      Cuts ← calcuateCuts (G(S, D), k)   ▹ Cuts may already have been calculated in a previous iteration.	
9:	      for all ((S′, D′), cutCost) ∈ Cuts do	
10:	       S″ ← S\S′, D″ ← D\D′	
11:	       ℳ′ ← ℳ \{(S, D)} ∪ {(S′, D′), (S″, D″)}	
12:	       T′ ← T where S is resolved by two nodes S′, S″	
13:	       cost′ ← cost + cutCost	
14:	       𝒫′ ← 𝒫′ ∪ {(ℳ′, T′, cost′)}	
15:	      end for	
16:	     end for	
17:	   end for	
18:	   𝒫 ← 𝒫′ reduced to the best k partial solutions	
19:	  end while	
20:	  return T1, …, Tk from 𝒫	
21:	 end function	
1:	function initPartialSolutions (G(S, D))	
2:	  ℳ ← {(S, D)}	
3:	  T ← tree with a single root S	
4:	  return {({(S, D)}, T, 0)}	
5:	end function	
1:	function calcuateCuts (G(S, D), k)	
2:	  if G(S, D) is disconnected then	
3:	    return {((S′, D′), 0)}	
4:	  else	
5:	    return {((S1′,D1′),c1),((S2′,D2′),c2),…,((Sk′,Dk′),ck)}   ▹ If cuts do not exist, calculate them with cut enumeration (Sec. 4.1
                             Enumeration) or cut sampling (Sec. 4.2).	
6:	      end if	
7:	end function	

Searching for Suboptimal Vertex-Cuts in G(S, D)

For the beam search algorithm described above, we have to compute k different vertex-cuts instead of a single minimum vertex-cut in the given graph G(S, D). This is achieved by computing suboptimal cuts in the network graph H(S, D′). In the following, we present two strategies for doing so; namely, suboptimal cut enumeration (Enum) and random cut sampling (CS).

Cut enumeration

Vazirani & Yannakakis (1992) introduced an algorithm to enumerate the cuts of a network in the order of non-decreasing weights, using O(|V|) maximum flow computations between two successive outputs. (A non-decreasing order allows that multiple solutions of identical weight can occur in the output.) Let H(S, D′) be the bipartite network with n = |S| taxon vertices, m = |D′| clade vertices and n ≤ m. Clearly, the network contains n + m vertices and O(nm) arcs. A maximum flow of H(S, D′) can be calculated in O(n2m) time by using the bipush variant of the preflow algorithm with dynamic trees, as described in Ahuja et al. (1994). Since we are only interested in enumerating cuts that separate the taxon set S, we only need n instead of n + m max flow computations, which leads to a running time of O(n3m) per minimum cut. We have to calculate at most k2 cuts in each of the O(n) partitions steps of BCD the algorithm, which leads us to the following lemma:

Lemma 2 Given an input matrix M over {0,1,?} for n taxa and m clades and an integer k ≥ 1, the BCD Beam Search algorithm using cut enumeration computes a supertree in O(k2mn4) time.

Hence, running time of this algorithm is still polynomial. The additional O(k2n) factor in the running time, compared to the original BCD algorithm, stems from using the Vazirani–Yannakakis algorithm. Running time may be reduced to O(k2mn3) if findings by Hao & Orlin (1994) for bipartite graphs can be adopted to the improved cut enumeration algorithm of Yeh, Wang & Su (2010).

As an algorithm engineering trick, we note that in practice, we usually have to compute much fewer cuts than the k2 mentioned above: We start by computing k cuts for the best partial solution; this gives us an upper bound for the k-th best cost in the active phase. Now, when we consider the second-best partial solution, we can stop as soon as the computed cost exceeds the upper bound; and we can update the upper bound in case we find partial solutions that belong to the top k.

Cut sampling

The CS algorithm is inspired by the randomized algorithm of (Karger & Stein, 1996) for finding all minimum cuts in an undirected graph U(S, E) with a certain probability. The algorithm recursively contracts edges of the graph and merges the connected vertices into sets, until only two vertex sets are left. Edges are randomly drawn with probability proportional to their edge weight. This contraction algorithm runs in O(|S|2) time and has Ω(1/|S|2) probability of outputting a minimum cut. The algorithm requires O(|S|2log|S|) iterations to find a minimum cut in one of the trials with high probability (probability converging to 1). Karger and Stein further presented a recursive contraction algorithm, where the trials share their work, so that each of the trials can be executed in O(1) time. The recursive contraction algorithm runs in O(|S|2log|S|) time, and outputs a minimum cut with probability Ω(1/log|S|). After O(log2|S|) iterations, the probability to find a minimum cut converges to 1, which results in an total running time of O(|S|2log3|S|).

Each of the trials returns a (potentially suboptimal) cut, and cuts with lower costs are sampled with higher probability than cuts with higher costs. To this end, the algorithm can also be used to generate a large number of suboptimal cuts without additional overhead: In detail, we can calculate O(|S|2log2|S|) cuts in O(|S|2log3|S|) time.

To apply the contraction idea here, we transform the bipartite graph G(S, D) into a simple undirected graph U = (S, E) with vertex set S. We insert edges to E so that for each c ∈ D, all t ∈ S adjacent to c form a clique in U. Let be the set of edges of the clique induced by c. Given the graph U, we choose the edge e ∈ E to contract in a two-step approach. First, we randomly pick a clade vertex c ∈ D with probability proportional to its weight. Second, we draw an edge e ∈ E(c) equally distributed, which is then contracted as described by Karger & Stein (1996). We contract edges until only two sets of vertices are left. These two vertex sets are a bipartition S′, S″ of our taxon set S, and the edges e ∈ E connecting S′ and S″ correspond to the clade vertices we need to delete in G(S, D) to induce this bipartition of S. See Fig. 1 for an exemplary workflow of the algorithm.

Figure 1 Workflow of the cut sampling algorithm (Section Cut Enumeration) including the transformation from G(S, D) to U(S, E) and the two-step edge selection process we use in the recursive contraction algorithm.

Edges E(c) ⊆ E are those edges with color (clade-vertex) c.

The above algorithm allows us to sample low-weight cuts with higher probability than high-weight cuts. When selecting the edges as described above, there exists for each cut in U(S, E) a vertex-cut in G(S, D) with the same partition and identical weight. With our modified edge selection process, we ensure that the probability of a clade vertex to be chosen in each contraction step is proportional to its weight. But clade vertices with higher degree are more likely to be deleted than clade vertices with lower degree and same weight. Therefore, the above algorithm has no guarantee to find a minimum cut with a certain probability, and we cannot guarantee that a minimum cut will be part of the output. But we can calculate a minimum cut in O(|D| |S|2) time using the maximum flow approach, and add it to the list of cuts.

The two-step approach needs O(|D|·|S|) time to choose and contract an edge, and O(|S|) contractions are needed to produce a cut. Using the recursive contraction algorithm, we need O(|D||S|2 log3|S|) time to calculate O(|S|2log2|S|) cuts. If we assume k ∈ O(|S|2log2|S|), which is realistic in practice, this leads us to the following lemma:

Lemma 3 Given an input matrix M over {0,1,?} for n taxa and m clades and an integer k ≥ 1 with k ∈ O(n2 log2 n), the BCD Beam Search algorithm using CS computes a supertree in O(kmn3log3n) time.

Again, we can do some algorithm engineering to improve running times in practice: We start by sampling cuts for the best partial solution; this again gives us an upper bound for the k-th best cost in the active phase. Now, we check if the score of the second-best partial solution plus the weight of an optimal cut (computed using the max flow approach) exceeds the upper bound; in this case, no sampling is required for this partial solution. We note that running time improvements by this trick are presumably smaller than for cut enumeration, as we have to run the full sampling process if the optimal cut does not result in a violation of the upper bound.

Experimental Setup

We evaluate the performance of the BCD Beam Search algorithm against the original BCD algorithm, MRP (Baum, 1992; Ragan, 1992), and SuperFine (Swenson et al., 2012). We concentrate on these methods as they have consistently outperformed other available supertree methods with respect to supertree accuracy, see (Swenson et al., 2010; Kupczok, Schmidt & von Haeseler, 2010; Swenson et al., 2011, 2012; Brinkmeyer, Griebel & Böcker, 2011, 2013; Fleischauer & Böcker, 2017) for evaluations. To allow for a fair comparison, we report results for several datasets (both simulated and biological) which have previously been used to access the performance of supertree methods. Evaluation on simulated data has the advantage that the true tree (also called model tree) is known and can be compared against; this is not the case for biological datasets. However, simulated data often have the disadvantage that the signal is “too strong,” and almost any method returns high quality results. Therefore, we have to ensure that findings made for simulated data are also supported for biological data.

Our evaluation closely follows that of the original BCD method; see (Fleischauer & Böcker, 2017) for details. MRP trees are computed using PAUP* (Swofford, 2002) and we report the majority consensus tree of all optimal trees. For SuperFine we use the implementation given by Swenson et al. (2012). BCD Beam Search also returns multiple trees; in such cases, we return the tree with the best BCD score. If multiple trees with identical score exist, we return the majority consensus of these trees. If no BS values are available for a dataset, BCD and BCD Beam Search are executed with UWs; if BS values are available, we additionally use BS weights. BCD and BCD Beam Search use the GSCM preprocessing described in Fleischauer & Böcker (2017).

Computations were executed on a single CPU core at 2.4 GHz and with 128 GB of memory.

Simulated data

We use the simulated SMIDGenOG datasets with 500 and 1,000 taxa (Swenson et al., 2010; Fleischauer & Böcker, 2016). Each replicate consists of multiple clade-based source trees using a densely sampled subset of taxa from one clade of the model tree, plus a single scaffold source tree which uses a sparsely-sampled subset of taxa of the complete model tree. The scaffold tree contains 25%, 50%, 75% or 100% of the taxa in the model tree (scaffold factor). A total of 30 replicate model trees is generated. For each replicate and each scaffold factor, we generate 15 (for 500 taxa) or 30 (for 1,000 taxa) clade-based source trees, plus a single scaffold tree with the desired scaffold factor. For further details, we refer the reader to Swenson et al. (2010) and Fleischauer & Böcker (2016, 2017).

Furthermore, we use the simulated SuperTriplets dataset (Ranwez, Criscuolo & Douzery, 2010). This dataset has only 100 taxa plus one outgroup. Here, the number of source trees varies between 10 and 50. Each set of source tree is available with 25%, 50% or 75% taxa deletion. For each configuration there exist 100 replicates.

Evaluation metrics for simulated data

To evaluate the quality of a reconstructed supertree on simulated data, it is common practice to evaluate splits (bipartitions on the set of taxa) that distinguish supertree and model tree. Despite our trees being rooted, we stick with this common practice. True positive (TP) splits are present in both the supertree and the model tree; false negative (FN) splits are not in the supertree but in the model tree; finally, false positive (FP) splits are in the supertree but not in the model tree. To evaluate supertree quality by a single number, we use the well-known F1 score, which is the harmonic mean of precision and recall: F1=21/precision+1/recall=2TP2TP+FP+FN

Biological data

We also evaluated the supertree methods using five biological supertree datasets, namely seabirds (121 taxa, seven source trees, see Kennedy & Page, 2002), temperate herbaceous tribes (THPL, 586 taxa, 22 source trees), see Wojciechowski et al. (2000), primates (85 taxa, 46 source trees, see Purvis, 1995), a mammalian phylogenomics case study (OMM, 33 taxa, 12,958 source trees, see Ranwez, Criscuolo & Douzery, 2010) and bats (916 taxa, 16 source trees), see Jones et al. (2002). See Table 1 for detailed information about the biological datasets. Note, the OMM and the bats dataset can be seen as extreme cases. The OMM dataset contains an enormous number of input trees compared to the number of taxa and is highly contradictory (96% conflicting splits). In contrast, the bats dataset contains no conflicts and thus allows for a perfect phylogeny.

Table 1 Overview of the biological datasets used in our evaluation.

Name	#Taxa	Input trees	
		#	Min	Max	Mean	Median	Conflicts (%)	BS	Root	
Primates	85	48	9	67	21.62	19.5	48.82	✓	✓	
Seabirds	129	7	15	90	32.14	22	23.71	✗	✓	
THPL	587	20	10	140	45.9	41.5	29.46	✗	✗	
OMM	33	12,958	6	33	27.8	29	96.25	✗	✓	
Bats	936	16	5	209	58.5	52.5	0	✗	✓	
Note:

Here, “#” is the number of input trees and BS indicated whether the input trees contain bootstrap values or not. Conflicts are the number of clades in the source trees that conflict with least one other clade from the source trees.

Evaluation metrics for biological data

For the biological datasets, the true tree is unknown but we can evaluate the supertree against the source trees. We measure the sum of false negatives (SFN) and sum of false positives (SFP) rate of a supertree T′ compared to the set source trees 𝒯, SFN rate=∑T∈T |S(T)∖S(T′|ℒ(T))|∑T∈T |S(T)| and SFP rate=∑T∈T |S(T′|ℒ(T))∖S(T)|∑T∈T |S(T′)|,

where T′|ℒ(T) is the subtree of T′ induced by the taxon set of tree T and 𝒮(T) is the set of splits induced by tree T. The optimal values of SFN rate and SFP and depend on the source trees; for conflicting source trees, no supertree can satisfy SFN rate = 0 and SFP rate = 0 simultaneously. For further information about this criterion see Fleischauer & Böcker (2017).

Results

We now describe results for different BCD Beam Search variants in comparison to BCD, MRP and SuperFine on simulated and biological data.

SMIDGenOG

The SMIDGenOG dataset contains BS values; to this end, we can evaluate BCD and BCD Beam Search using the BS weighting. On this dataset, BCD with UWs is already on par with MRP and SuperFine; BCD with BS weights outperforms MRP and SuperFine. We ran BCD Beam Search with k = 25 partial solutions. We find that BCD Beam Search with BS weights consistently outperforms any other evaluated methods with respect to F1 (see Figs. 2A and 2B); this is true for both the 500 and 1,000 taxa dataset. On the 500 taxa dataset with 120 instances, BCD Beam Search with cut enumeration outperforms BCD on 83 instance (19 ties), SuperFine on 111 instances (1 tie) and MRP on 113 instances (1 tie). When using CS it outperforms BCD on 68 instances (32 ties), SuperFine on 111 instances (1 tie) and MRP on 113 instances. On the 1,000 taxa dataset with 120 instances, BCD Beam Search outperforms BCD on 100 instance (seven ties) when using cut enumeration and on 94 instances (17 ties) when using CS. Both BCD Beam Search variants outperform MRP on 119 and SuperFine on all 120 instances.

Figure 2 F1 score (A and B) and running times (C and D) of SuperFine, MRP, BCD, and BCD Beam Search variants on the simulated SMIDGenOG dataset. Results for the 500 taxa dataset (A and C) and the 1,000 taxa dataset (B and D).

The x-axis shows different scaffold factors in percent, see Swenson et al. (2010) and Fleischauer & Böcker (2016, 2017) for details.

We report running times for the SMIDGenOG dataset in Figs. 2C and 2D. We see that BCD Beam Search with cut enumeration is about two- (500 taxa) to three-fold (1,000 taxa) slower than SuperFine. Whereas BCD Beam Search with CS is slightly slower than SuperFine on the 500 taxa dataset, it is already slightly faster on the 1,000 taxa dataset. As expected, the beam search is slower than the regular BCD algorithm; namely up to 20-fold slower for cut enumeration and fivefold slower for CS. But notably, it is on average 10-/15-fold faster than MRP (500/1,000 taxa) for cut enumeration and 15-/46-fold faster than MRP (500/1,000 taxa) for CS. For the 1,000 taxa dataset, the average running time of BCD is less than 6 s. BCD Beam Search with cut enumeration needs less than 2 min; BCD Beam Search with CS and SuperFine need less than 1 min; and MRP needs about 27 min.

The number of suboptimal solutions (k) shows a quadratic impact on the running time for the beam search with cut enumeration, whereas the running time increases only linear for beam search with CS (see Fig. 3). Further, we found that even for k = 100 the beam search with CS is still less than twofold slower than SuperFine and still clearly faster than MRP. With k = 100 the beam search with cut enumeration is always faster than MRP.

Figure 3 Running times of BCD Beam Search with different numbers of suboptimal solutions (k = 1, k = 25, k = 50, k = 75, and k = 100) on the simulated SMIDGenOG dataset.

Results for the 500 taxa dataset (A) and the 1,000 taxa dataset (B) The x-axis shows different scaffold factors in percent.

SuperTriplets Benchmark

This dataset does not contain BS values or BLs; to this end, BCD has to be run with UWs and, hence, often showed worse accuracy than MRP and SuperFine in previous evaluations (Fleischauer & Böcker, 2017). We ran BCD Beam Search with k = 25 and k = 50 partial solutions (see Fig. 4). It is still the case that BCD supertrees are generally of lower quality than MRP and SuperFine supertrees: For 10 input trees, BCD Beam Search performs on par with MRP for deletion rate 25%; for 50 input trees, it is on par with MRP for deletion rates 25% and 50%. For the remaining configuration it performs worse than MRP; this is particularly the case for 50 input trees and 75% taxa deletion. But we observe that BCD Beam Search reaches a significantly higher F1 than BCD, for all numbers of input trees and taxa deletion rates. In contrast, we do not observe a significant increase of F1 when considering k = 50 instead of k = 25 partial solutions.

Figure 4 Score of SuperFine, MRP, BCD, and BCD Beam Search variants on the simulated SuperTriplets Benchmark.

Results for 10 input trees (A) and 50 input trees (B). The x-axis shows different data deletion rates within each source tree, see Ranwez, Criscuolo & Douzery (2010) and Fleischauer & Böcker (2017) for details.

Bad Clade Deletion Beam Search with cut enumeration and k = 25 partial solutions produced a supertree with higher F1 than BCD without beam search for 1,212 of 1,500 replicates; of the remaining, 144 are ties. This is very similar for BCD Beam Search with CS, with 1,191 wins and 146 ties. We stress that many replicates resulted in supertrees with identical BCD Score, indicating the combinatorial complexity of this dataset.

Biological data

Bad Clade Deletion Beam Search supertrees show significantly better (lower) SFN rates and SFN rates on all biological datasets than the original BCD, see Table 2: We find that best SFN rates/SFP rates are distributed between MRP, SuperFine, and BCD Beam Search, but no method constantly outperforms the others over all datasets. This is remarkable, as none of the datasets but the primates data contain BS values or BLs, and BCD Beam Search with UWs had to be applied. Recall, the bats data allow for a perfect phylogeny. The results for this data show that BCD always finds a perfect phylogeny if one exists whereas MRP may not. Due to the stochastic nature of the CS procedure, we see that BCD Beam Search results differ between cut enumeration and CS; the cut enumeration approach tends to be more robust, as expected.

Table 2 Sum of false negative rates and the sum of false positive rates of supertrees against source trees on biological supertree datasets.

Dataset	SFN rate/SFP rate	
	Primates	Seabirds	THPL	OMM	Bats	
MRP	0.169/0.165	0.153/0.159	0.190/0.328	0.386/0.425	0.063/0.015	
SuperFine	0.165/0.172	0.127/0.206	n/a	n/a	n/a	
BCD UW	0.178/0.185	0.153/0.206	0.454/0.523	0.425/0.457	0/0	
BCD BS	0.174/0.180	n/a	n/a	n/a	0/0	
BCD-K25-Enum UW	0.172/0.178	0.122/0.175	0.239/0.320	0.386/0.417	0/0	
BCD-K25-Enum BS	0.165/0.169	n/a	n/a	n/a	0/0	
BCD-K25-CS UW	0.176/0.183	0.122/0.175	0.250/0.334	0.388/0.420	0/0	
BCD-K25-CS BS	0.169/0.176	n/a	n/a	n/a	0/0	
Note:

Most datasets do not contain bootstrap values, prohibiting the use of Bootstrap weights. Best rates in each column are marked in bold. We do not have results for SuperFine on OMM because the GSCM did not finish in reasonable time. Since the GSCM tree of the OMM data does not contain a single clade (calculated with BCD), SuperFine results are identical to MRP anyways. For Bats and THPL, SuperFine did not return a result due to too less overlap of the input trees in the unrooted case.

Conclusion

We presented a beam search algorithm that allows the BCD algorithm to consider the k best partial solutions instead of only the optimal one, when partitioning the taxon set in a top-down manner. BCD Beam Search has still guaranteed polynomial running time. We introduced an algorithm to enumerate suboptimal solutions in non-decreasing order, and a second algorithm to sample good partial solutions. Our evaluations on simulated and biological data showed that both beam search approaches consistently improve BCD on all evaluated datasets for k ≥ 25. Both methods for computing suboptimal cuts perform roughly on par, but the enumeration algorithm tends to be more robust. However, the sampling algorithm scales linearly with the number of suboptimal solutions to be considered, whereas the exact enumeration algorithm scales quadratically in the worst case. We further found that BCD Beam Search, especially when used together with BS weights, is on par with MRP and SuperFine even on biological data with regards to supertree quality. This has not been achieved previously by any worst-case polynomial time supertree method. Finally, BCD can significantly outperform MRP and SuperFine for very large datasets with several thousand taxa (Fleischauer & Böcker, 2017).

Availability

The BCD Beam Search algorithm has been implemented in Java as part of the open source BCD command line tool (since v1.1) which is publicly available on GitHub (https://github.com/boecker-lab/bcd-supertrees). All data used from the evaluation are publicly available online (https://doi.org/10.6084/m9.figshare.6189113).

Additional Information and Declarations

Competing Interests

Author Contributions

Data Availability

The authors declare that they have no competing interests.

Markus Fleischauer conceived and designed the experiments, performed the experiments, analyzed the data, contributed reagents/materials/analysis tools, prepared figures and/or tables, authored or reviewed drafts of the paper, approved the final draft.

Sebastian Böcker conceived and designed the experiments, contributed reagents/materials/analysis tools, authored or reviewed drafts of the paper, approved the final draft.

The following information was supplied regarding data availability:

Source code at GitHub: https://github.com/boecker-lab/bcd-supertrees.

Fleischauer, Markus (2018): BCD Beam Search evaluation data. figshare. Fileset. https://doi.org/10.6084/m9.figshare.6189113.v1.

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
