# Peer review of "BCD Beam Search: considering suboptimal partial solutions in Bad Clade Deletion supertrees"

_PeerJ, doi:10.7717/peerj.4987_

## Round 0.1 · original submission · Minor Revisions

The authors are encouraged to submit a revised paper by carefully considering the comments of the reviewers.

Reviewer 1 ·

Basic reporting

Authors present an improvement of their previously introduced Bad Clade Deletion (BCD) algorithm, a polynomial greedy supertree method. Supertree methods, which consist to assemble partial phylogenies to a large and unique one while minimizing conflicts, constitute a major and widely studied problem in phylogenetics. BCD minimizes the number of matrix columns deletions to resolve the input matrix conflicts (the input matrix contains cladistics information of all input trees). Here, the paper presents a beam search algorithm (BCD BS) that improves the efficiency of BCD. At each step of the supertree construction (refinement), BCS BS considers simultaneously k partial solutions instead of 1. Naturally, reconstructed supertrees are (slightly) better with the beam stream technique (with k=25 in experiments) than without, while running times increase linearly according to k.

The paper is globally clear, well written, and adequately structured. Algorithms are sufficiently detailed (but not necessarily in the best way, see below) and experimentations perfectly show the efficiency of the proposed mechanism.

However it is certainly possible to significantly improve the description of the problem and especially the methods by adding figures and algorithms. This would provide a better understanding of the contribution, especially for a reader who is not already familiar with MCD and BCD. As is stands, the article is quite short but dense in technical descriptions.
In the phylogenetic trees preliminaries, some definitions seems not perfectly clear (induced subtree, parent tree, lines 84-87).

Very few typos noted:
- Abstract: both beam -> Both beam
- line 216: solution -> solutions
- line 396: Table ?? -> Table 1

Experimental design

see (1)

Validity of the findings

Good.

·

Basic reporting

Everything here is basically ok. The MS is well structured and clearly written and it is easy to follow what the authors have done and why they have done it. The results are also presented clearly and accurately.

The English does need a bit of polishing, however, especially later on in the MS. Nothing major and nothing that detracted from the article, but a bit of spit and polish would be good.

Experimental design

This is an interesting MS that makes a useful advance toward supertree methodologies. The problem with almost any (meaningful) phylogenetic analyses is that heuristic searches have to be used to get around the NP-complete nature of the problem. Therefore, any polynomial time solutions are most welcome!

I can't really dissect the math behind the new method (so a review from a mathematician is definitely needed), but I can say that the testing of their method with both simulated and empirical data is robust and shows the performance of the different methods under a range of conditions. Thus, I am confident that the comparative results are accurate.

Validity of the findings

For the most part, see my comments above. The experimental design is thorough and well performed such that it is my opinion that the results regarding the comparative performance of the different methods are accurate.

Additional comments

Some more minor and /or specific comments that I had:

L13: It's not really true that differing process can lead to conflicts because even the same one (e.g., incomplete lineage sorting, ILS) can cause gene trees to disagree with one another and still be accurately for the evolutionary history of that gene. On the flip side, different evolutionary processes (e.g., ILS and horizontal gene transfer) could, by chance, lead to the same result.

L48: But was the quality any good per se? It might be the best result, but still not a very good one.

L57: State-of-the-art tends to mean the newest, which MRP definitely isn't. Perhaps "established"?

L61: Add an "s" to "branch length".

L88: "taxa set" should be "taxon set", here and throughout.

L98- Spell out "1-1".

L186: "partial solutions".

L210: "two graphs have"

L230: Explain what "non-decreasing weights" are? I take it that this is something different than simply increasing weights?

L236: Delete the word "do".

L264: log to the power of 3???

L281: Is such a guarantee even possible? OR does the problem fall into the NP-complete/hard class such that it might be possible.

L282: What does "high probability" mean here exactly? Rather vague.

L309: Specify that it is outperformed WRT accuracy.

L314: Simulated data, however, often have the disadvantage that the signal is too strong such that almost any method does well.

L346ish: If the true tree is unknown, how can one measure the false positives or negatives? Explain.

L353: Some rough numbers would be helpful in this paragraph. For instance, can one say that the performance increase was usually two-fold better (or something like that)? (As in the next paragraph.)

L364: Although the two-fold increases are mentioned here, some real numbers would be good too. A two-fold increase on a running time of 10 seconds is trivial, but not for 10 days!

L368: "As expected, the beam search is".

L373: "whereas the running time increases only linearly for".

L380: Define "suboptimal".

L382: So it would appear that BCD requires some form of branch length or (bootstrap) support?

L395: Improves WRT what?

L396: Add the Table number.

L398 Any idea why? Have you tried to see how much conflict there is in the data set? If there is very little then perhaps all methods point to the same (obvious) solution.

---

## Round 0.2 · accepted · Accept

The revised version incorporated the suggestions from the reviewers.

# Reviewer 1 ·

Basic reporting

The authors fully took into account the remarks made after the first submission. The article was already nice and very pleasant to read. Complementary figure and pseudo-code make the contribution and algorithm easier to understand. New Table 1 and extreme cases instances are also judicious additions.

Experimental design

Rigorous and replicable study. The experimental setup is well described. The Java code is provided and publicly available.

Validity of the findings

The efficiency of BCD Beam Search is experimentally assessed, and conclusions are supported by appropriate experiments.

Additional comments

Minor remarks and typos:
- line 361: "bats" should be emphasized, and "see Jones et al. 2002" included in parentheses
- line 381: instanceS

·

Basic reporting

Fine.

Experimental design

Improved with the additional explanation and the added empirical samples.

Validity of the findings

Again, improved with the added empirical examples.

Additional comments

The revisions to the paper, although relatively minor, have strengthened what was already a good paper describing an interesting supertree method and I recommend that the paper be published as is.